# *β*-Cyclocitral from *Lavandula angustifolia* Mill. Exerts Anti-Aging Effects on Yeasts and Mammalian Cells via Telomere Protection, Antioxidative Stress, and Autophagy Activation

**DOI:** 10.3390/antiox13060715

**Published:** 2024-06-12

**Authors:** Jiaheng Shan, Jianxia Mo, Chenyue An, Lan Xiang, Jianhua Qi

**Affiliations:** College of Pharmaceutical Sciences, Zhejiang University, Yu Hang Tang Road 866, Hangzhou 310058, China; 22119073@zju.edu.cn (J.S.); mojx@zju.edu.cn (J.M.); 22119059@zju.edu.cn (C.A.)

**Keywords:** *Lavandula angustifolia* Mill., *β*-cyclocitral, anti-aging, telomere protection, antioxidative stress, autophagy

## Abstract

We used a replicative lifespan (RLS) experiment of K6001 yeast to screen for anti-aging compounds within lavender extract (*Lavandula angustifolia* Mill.), leading to the discovery of *β*-cyclocitral (CYC) as a potential anti-aging compound. Concurrently, the chronological lifespan (CLS) of YOM36 yeast and mammalian cells confirmed the anti-aging effect of CYC. This molecule extended the yeast lifespan and inhibited etoposide (ETO)-induced cell senescence. To understand the mechanism of CYC, we analyzed its effects on telomeres, oxidative stress, and autophagy. CYC administration resulted in notable increases in the telomerase content, telomere length, and the expression of the telomeric shelterin protein components telomeric-repeat binding factor 2 (TRF2) and repressor activator protein 1 (RAP1). More interestingly, CYC reversed H_2_O_2_-induced telomere damage and exhibited strong antioxidant capacity. Moreover, CYC improved the survival rate of BY4741 yeast under oxidative stress induced by 6.2 mM H_2_O_2_, increasing the antioxidant enzyme activity while reducing the reactive oxygen species (ROS), reactive nitrogen species (RNS), and malondialdehyde (MDA) levels. Additionally, CYC enhanced autophagic flux and free green fluorescent protein (GFP) expression in the YOM38-*GFP-ATG8* yeast strain. However, CYC did not extend the RLS of K6001 yeast mutants, such as Δ*sod1*, Δ*sod2*, Δ*cat*, Δ*gpx*, Δ*atg2*, and Δ*atg32*, which lack antioxidant enzymes or autophagy-related genes. These findings reveal that CYC acts as an anti-aging agent by modifying telomeres, oxidative stress, and autophagy. It is a promising compound with potential anti-aging effects and warrants further study.

## 1. Introduction

The accelerating process of global population aging and the increasing burden of age-associated chronic diseases necessitate attention [1]. Common age-associated chronic diseases, including cardiovascular disease, type 2 diabetes, metabolic syndrome, and some cancers, present significant challenges for elderly patients, making the concept of healthy aging crucial. Healthy aging aims to not only extend lifespan but also maintain functional ability and overall well-being in older individuals [2]. It is widely recognized that adopting healthy lifestyles can prevent age-related chronic illnesses and promote healthy aging. However, obstacles such as unhealthy work demands and limited personal motivation for sustained behavioral changes often impede these efforts [1,2]. Consequently, biomedical approaches for disease prevention now focus on developing pharmaceutical interventions to regulate the aging process itself.

Comprehending the mechanism behind aging provides assistance in the development of precision-targeted pharmaceutical interventions. Aging is associated with molecular pathways such as telomere attrition, oxidative stress, autophagy dysregulation, genomic instability, stem cell depletion, and epigenetic changes [3]. Telomere attrition, in particular, is widely recognized as a crucial factor in senility, as it triggers the various aging characteristics mentioned above. Significant telomere shortening leads to cellular senescence or division arrest, which is thought to be the underlying cause of many chronic diseases [4]. Telomerase, an enzyme that utilizes an RNA template to incorporate telomeric DNA into telomeres, helps counteract the wear and tear resulting from cell division [5]. Telomeres, which are specialized structures consisting of DNA tandem repeats with specific proteins, prevent improper recombination between telomeres and chromosome end-to-end fusions [6]. In vertebrates, these specialized proteins are referred to as shelterin proteins, with telomeric-repeat binding factor 2 (TRF2) and repressor activator protein 1 (RAP1) being essential components [4]. TRF2 directly binds to double-stranded telomeric repeats, and its deficiency promotes cellular senescence and apoptosis in patients with Behcet’s disease [7]. RAP1, on the other hand, is recruited to telomeres by TRF2 and protects against obesity and insulin resistance in mice [8]. Previous studies have shown that adult mammalian tissues do not possess sufficient telomerase to maintain telomere length throughout the lifespan [3,4,5,6,7,8]. Therefore, it may be beneficial to activate telomerase, such as astragaloside IV (AST), and enhance the expression of telomeric regulatory proteins using external methods to preserve telomere integrity [9].

Identifying the underlying causes of variation in telomere attrition may provide crucial insights into the processes of aging and their connection with the environment. Oxidative stress is the result of the inevitable production of reactive oxygen species (ROS) during energy synthesis by the mitochondria, which is thought to be one of the main drivers of telomere attrition [10]. Telomeres are preferential targets for ROS as they contain many guanine base pairs prone to oxidation [10]. Additionally, ROS binds to nitric oxide (NO) to produce reactive nitrogen species (RNS) in organisms, triggering nitrosation stress, which leads to the hydroxylation of aromatic rings of amino acid residues, ultimately causing apoptosis or cell damage [11]. ROS and RNS are the two primary categories of free radicals, whose buildup is believed to play a significant role in cellular aging and the pathophysiology of diseases [12]. Endogenous antioxidant defenses, such as superoxide dismutase (SOD), catalase (CAT), and glutathione peroxidase (GPx), can maintain the balance between oxidative and antioxidative processes [13]. However, in chronic oxidative stress, the effectiveness of endogenous antioxidants is inadequate [10,11,12,13], underscoring the potential role of exogenous antioxidants such as resveratrol (RES) [14], taurine (TAU) [15], and *β*-carotene [16] in preserving homeostasis.

In addition to the endogenous redox system, organisms have other molecular signaling pathways to resist external stress, such as autophagy. It is a cellular degradation process reliant on lysosomal mechanisms, which removes redundant or dysfunctional cellular components [17]. It has been shown to eliminate mitochondria, the endoplasmic reticulum, peroxisomes, and proteins damaged by oxidative stress, offering therapeutic potential in various diseases, including metabolic disorders, neurodegenerative diseases, infections, and cancer [17]. Therapeutic strategies utilizing autophagy enhancers such as rapamycin (RAP) have shown promising results in animal models of Parkinson’s disease, which are characterized by reduced α-synuclein accumulation, oxidative damage, dysfunctional mitochondria, and neurological harm [18]. Moreover, increasing autophagy promotes longevity and delays the aging process through caloric restriction and dietary deprivation [19]. Additionally, enhancing autophagy restores the regenerative capacity of senescent stem cells [20]. As a result, interventions that enhance autophagy may offer hope for promoting successful aging and longevity.

During an aging study, it is most important to select the suitable organism as the aging model. *Saccharomyces cerevisiae* has emerged as a favored model for geriatric research due to its short life cycle, affordability, and versatility in molecular biology manipulations [21]. Observations have revealed that the mortality curve of yeast populations resembles that of various organisms, including humans. As a result, the distinct lifespan curves of yeast, namely, the replicative lifespan (RLS) and chronological lifespan (CLS), present a valuable opportunity to investigate aging in both dividing and nondividing cells [21,22,23]. In addition to yeast, a variety of mammalian cell models have been employed in anti-aging research, highlighting the broad applicability of anti-aging substances [24,25]. For example, *β*-coumarin significantly decreased the death rate of the rat adrenal pheochromocytoma cell line PC12 induced by amyloid-beta 42 and significantly extended the lifespan of Alzheimer’s disease-affected flies [24]. Ginseng oligopeptides delay oxidative stress-induced senescence of mouse embryonic fibroblast line 3T3 cells through their antioxidant and anti-inflammatory capacities [25]. In our study, the diverse properties of yeast made it an ideal candidate for large-scale anti-aging drug screening and mammalian cells help us validate their anti-aging mechanisms efficiently.

Natural products are important resources for the research and development of drugs. In the pre-experiment, we found that *L. angustifolia* Mill. is an anti-aging active material from a variety of natural products with a bioassay system. *L. angustifolia* Mill. is native to the southern Alps along the Mediterranean coast and is a renowned herb that is cultivated in the Xinjiang Uygur Autonomous Region, China [26,27]. Lavender extract (derived from *L. angustifolia* Mill.) is a complex mixture of more than 100 aromatic compounds. Notable constituents include linalool, linalool acetate, eucalyptus, lavender acetate, lavender alcohol, terpene-4-ol, and camphor [26]. To date, lavender extract has been found to possess antioxidant, calming and hypnotic, cognitive-enhancing, antibacterial, anti-inflammatory, and antihypertensive effects [28]. The anti-aging properties of the plant’s volatile extract have also attracted attention, and these have been widely incorporated into cosmetics and daily goods in recent years [29]. Nevertheless, the underlying active substances responsible for its anti-aging effects, particularly small molecule compounds, and their corresponding mechanisms of action have yet to be fully elucidated.

In this study, we utilized the K6001 yeast replicative lifespan biological activity system as a guide to isolate lavender extract. We then separated and purified several volatile compounds and found that *β*-cyclocitral (CYC) exhibited the most significant activity in extending the lifespan of yeast. This finding demonstrated the strong potential of CYC in delaying senescence. However, little research has been conducted on the anti-aging activity of CYC, and its mechanism of action remains unclear. In this paper, we present evidence that CYC exerts an anti-aging effect on both yeast and mammalian cells by activating telomerase, telomere length, and the telomeric regulatory proteins TRF2 and RAP1. Additionally, CYC enhances oxidative stress resistance and induces autophagy.

## 2. Materials and Methods

### 2.1. General

Analytical-grade reagents, including *n*-pentane, *n*-hexane, methylene chloride, methanol, ethanol, paraformaldehyde, chloroform, and hydrogen peroxide from Sinopharm Chemical Reagent Co., Ltd. (Shanghai, China), were used in this study. The following substances and reagents were acquired from the listed vendors: RES (J&K Scientific Ltd., Beijing, China); etoposide (ETO) and RAP (Solarbio, Beijing, China); TAU and AST (Yuanye Bio-Technology Co., Ltd., Shanghai, China); 4′,6-diamidino-2-phenylindole (DAPI) (Macklin, Shanghai, China); dimethyl sulfoxide (DMSO) (Sigma, Saint Louis, MO, USA). Ethanol was used as a solvent for compound dissolution or as a negative control in yeast-related assays, whereas DMSO played the same role in PC12 and 3T3 cell experiments. Other common medium and reagent configurations are described in Appendix A.

### 2.2. Isolation and Purification of β-Cyclocitral

*L. angustifolia* Mill. was purchased from Integria Healthcare Pty Ltd., Eight Mile Plains, QLD, Australia. Three hundred grams of dried lavender flowers was distilled, and three grams of lavender extract was collected after condensation. Three grams of lavender extract was chromatographed on a silica gel column eluted with *n*-hex/CH_2_Cl_2_ (100:0, 90:10, 80:20, 70:30, 60:40, 0:100) to obtain eight fractions from **I-A-1** to **I-A-8**. All the fractions were tested for K6001 replicative lifespan activity, and the most active fraction, **I-A-6** (100.0 mg), was eluted with *n*-hex/CH_2_Cl_2_ (70:30). Fraction **I-A-6** (10.2 mg) was subjected to HPLC purification using a Cosmosil 5C18-MS-II packed column (*Φ*10/250 mm) with MeOH/H_2_O in a linear gradient of 40:60–100:0 over 80 min. The detection wavelength was set at 210 nm, and the flow rate was 3 mL/min. This purification process resulted in the isolation of a molecule, **I-4** (2.8 mg, t_R_ = 40 min). This molecule, **I-4**, was identified as *β*-cyclocitral by comparing the MS and ^1^H NMR spectral data with the previous literature [30,31]. HR ESI-TOF-MS m/z [M+H]^+^ 153.1217; ^1^H NMR (500 MHz, Chloroform-d) Δ 10.13 (s, 1H), 2.19 (t, *J* = 6.3 Hz, 2H), 2.09 (s, 3H), 1.67–1.58 (m, 2H), 1.47–1.41 (m, 2H), 1.20 (s, 6H). The chemical structure of CYC is shown in Figure 1a. The specific separation procedure and conditions can be found in Appendix A.

### 2.3. Yeast and Mammalian Cell Strains

The K6001 yeast strain was provided by Professor Michael Breitenbach (University of Salzburg, Salzburg, Austria). The K6001 strains Δ*sod1*, Δ*sod2*, Δ*cat*, Δ*gpx*, Δ*atg2,* and Δ*atg32*; BY4741; YOM36 and YOM38 containing the pR316-*GFP-ATG8* plasmid were obtained from Professor Akira Matsuura (Chiba University, Chiba, Japan). S288C yeast was purchased from Biosci Plasmid Strain Resources, Inc. (Hangzhou, China). The genotypes of all of the above yeast strains are listed in Appendix A. PC12 cells were obtained from the National Collection of Authenticated Cell Culture (Shanghai, China). 3T3 cells were purchased from MeilunBio Technology Co., Ltd., (Dalian, China).

### 2.4. The Replicative and Chronological Lifespan in Yeast

The replicative and chronological lifespan assays were conducted following the methodology outlined in a previous investigation [32]. For the current investigation, the anti-aging impact of CYC at concentrations of 0, 0.03, 0.1, 1, 3, 10, and 30 μM was assessed using the replicative lifespan of K6001 yeast. In brief, approximately four thousand yeast cells were spread onto peptone dextrose (YPD) agar plates containing samples. After being incubated for 48 h at 28 °C, forty microcolonies were chosen randomly from the plates, and the count of daughter cells produced by a single mother cell was recorded. The replicative lifespan assay of Δ*sod1*, Δ*sod2*, Δ*cat*, Δ*gpx*, Δ*atg2,* and Δ*atg32* yeast mutants with a K6001 background was determined as above.

The anti-aging effect of YOM36 at concentrations of 0.1, 1, and 10 μM was confirmed on the basis of its chronological lifespan [32]. In short, YOM36 yeast cells (OD_600_ = 0.01) were dispersed into synthetic defined (SD) medium containing samples for 72 h incubation. Subsequently, about two hundred yeast cells were spread onto YPD agar plates for 48 h incubation and the number of surviving colony-forming units (CFUs) were recorded. This process was repeated every two days until the survival rate (calculated as CFUs divided by CFUs on day 3, multiplied by 100%) fell below 5%. Detailed information about the lifespan experiments is provided in the Appendix A.

### 2.5. Observation of Senescence-Associated β-Galactosidase (SA-β-Gal) in Mammalian Cells

To evaluate SA-*β*-Gal activity, approximately 50,000 PC12 or 3T3 cells were seeded in each well of a 24-well plate and incubated for 24 h at 37 °C with 5% CO_2_. Subsequently, various test samples were applied to the cells. RAP served as the positive control, whereas 0.5% DMSO served as the negative control. After one day, the PC12 cells were treated with 1 µM ETO, and the 3T3 cells were treated with 0.3 µM ETO for 2 days. SA-*β*-Gal activity was then assessed using a senescence *β*-galactosidase staining kit (Beyotime Biotechnology, Shanghai, China) following the manufacturer’s instructions. The stained cells were examined using a bright-field microscope (Olympus Corporation, Tokyo, Japan), and ImageJ software (Version 1.42q, National Institutes of Health, Rockville, MD, USA) was used to quantify the percentage of SA-*β*-Gal. Three regions were measured for each group, with more than 150 cells counted per region.

### 2.6. Evaluation of Telomerase Contents in Yeast

The S288C yeast strain was inoculated in glucose liquid media for 24 h. Afterwards, approximately 30 million yeast cells were transferred to glucose liquid media and treated with 10 µΜ RES or 0, 0.1, 1, or 10 µM CYC for 48 h incubation at 28 °C. Subsequently, yeasts from each group were collected into 1 mL tubes. An amount of 500 µL PBS and grinding beads (Shanghai Jingxin Inc., Shanghai, China) were added into the yeasts in each tube. The mixtures were sonicated and centrifuged to obtain protein extraction. Protein concentrations were determined using a BCA kit (CoWin Biotech, Beijing, China), and each sample was diluted to 10 µg/µL.

The telomerase content of the yeasts was assessed using an enzyme-linked immunoassay kit purchased from Tongwei, Shanghai, China. The prepared microtiter wells containing the telomerase antibody were sequentially filled with 50 µL blank (sample diluent), 50 µL treated-group protein supernatant (20 µL protein solution plus 30 µL sample diluent), or 50 µL standards with different concentrations (12.5, 25, 50, 100, or 200 U/L) and incubated for 30 min at room temperature. After that, the liquid in each well was removed and washed by scrubbing solution five times. Subsequently, the 50 µL horseradish peroxidase-labelled detection antibody was added into each well, respectively. All of the wells that were incubated in a warm bath caused a color change from blue to yellow, and the intensity of the color was measured at 450 nm using a BioTek microplate reader (BioTek, Winooski, VT, USA). The standard curve, created by plotting the average optical density obtained for six standard concentrations on the vertical (Y) axis, was used to determine the telomerase amount in the samples. The standard curve equation for the telomerase content, y = 518.0 × −0.8821 (R^2^ = 0.9916), indicated good feasibility and linearity. Moreover, a positive correlation was observed between the plant telomerase content and color intensity.

### 2.7. DNA Extraction and Measurement of DNA Quality

About two million 3T3 cells were seeded in 10 mL CM sodium and incubated with 10 µL AST or 0, 0.1, 1, and 10 µL CYC for 48 h at 37 °C. Genomic DNA (gDNA) was extracted from the cultivated cells using the Mammalian Genomic DNA Extraction Kit (#D0061, Beyotime, Shanghai, China). In brief, 3T3 cells were incubated by 500 μL of the lysate sample containing proteinase K in a 50 °C water bath overnight. After full digestion of cells, an amount of 500 μL Tris-equilibrated phenol and chloroform was added into the cell lysate to extract gDNA three times, respectively. Subsequently, approximately 300 μL of the supernatant was aspirated and mixed with 60 μL of 10 M ammonium acetate and 600 μL of absolute ethanol to precipitate the DNA, which will form a white flocculent. The resulting DNA pellet was washed with 75% ethanol, then the ethanol was allowed to evaporate at room temperature to obtain purified gDNA. The NanoDrop 2000 microspectrophotometer (Thermo Fisher Scientific Inc., Waltham, MA, USA) was used to measure the gDNA purity (A_260_/A_280_ ratio) and concentration (ng/µL). Agarose gel electrophoresis, with an agarose gel consisting of agarose (Beijing, Shanghai, China), 1 × TAE solution (Solarbio, Beijing, China), and Dnred nucleic acid dye (Brigen, Beijing, China), was employed to assess gDNA purity. The *λ*-Hind III digest (Takara Biomedical Technology Co., Ltd., Beijing, China) was used as the DNA maker. Subsequently, the samples were added to the gel, followed by electrophoresis using a JEOL JMS-700 MStation spectrometer (JEOL, Tokyo, Japan) for 35 min at 135 V. The visualization of DNA bands enabled the assessment of gDNA integrity.

### 2.8. Calculation of Relative Telomere Length

To calculate the relative telomere length, the Relative Human Telomere Length Quantification qPCR Assay Kit (ScienCell, Research Laboratories, CA, USA) was utilized. This kit allows for the direct comparison of average telomere lengths between samples. A region of 100 bp on mouse chromosome 10, recognized and amplified by the single-copy reference primer set, served as a reference for data normalization. Following the manufacturer’s instructions, a reaction containing two nanograms of target DNA, two pairs of primers (telomere or SCR), and 2X GoldNStart TaqGreen qPCR Master mix was added to a total reaction volume of 20 µL. The PCR experiments were performed on a LightCycler^®^ 480II Real-Time PCR Instrument (Roche, Basel, Switzerland) and analyzed using LightCycler^®^ 480II software (Version 1.5.0.39, Roche, Basel, Switzerland). The SCR and telomere reactions were conducted in the same 96-well plate using the same qPCR program arrangement as specified in Appendix A. Subsequently, the Cq values of each group were obtained, and the relative telomere length was calculated according to the provided formula: ΔCq (SCR) = Cq (Sample, SCR) − Cq (Control, SCR); ΔCq (TEL) = Cq (Sample, TEL) − Cq (Control, TEL); ΔΔCq = ΔCq (TEL) − ΔCq (SCR); relative telomere length (Sample to Control) = 2^−ΔΔCq^.

### 2.9. Viability of 3T3 Cell under H_2_O_2_ Treatment

The concentration of H_2_O_2_ required to induce oxidative stress was determined via a cell viability assay following incubation with 3-(4,5-dimethyl-2-thiazolyl)-2,5-diphenyl tetrazolium bromide (MTT). The cells were initially incubated with blank EM solution for 48 h, followed by incubation with EM solution containing varying concentrations of 0, 0.2, 0.4, 0.6, 0.8, 1.0, 1.2, and 1.4 µM H_2_O_2_ for 2 h. Each well was then treated with 0.5 mL of fresh medium containing 200 µg/mL MTT for an additional two hours at 37 °C after removing the previous medium. The formaldehyde concentration was measured at an absorption value of 570 nm in each well.

### 2.10. Antioxidative Capacity of CYC in Yeast and Mutants

The antioxidative capacity of CYC was assessed using a previously established methodology [32]. Yeast resilience to oxidative stress under 0.1, 1, and 10 µM CYC treatments was evaluated based on yeast growth visualization and survival rate. Additional details of the experiment can be found in the Appendix A. The free radical ROS and malondialdehyde (MDA) levels were measured at CYC concentrations of 0.1, 1, and 1 µM using an ROS assay kit (Beyotime Biotech, Shanghai, China) and an MDA assay kit (Nanjing Jiancheng Bioengineering Institute, Nanjing, China), respectively. Further information regarding these experiments is provided in the Appendix A. Alterations in the activity of the antioxidant enzymes SOD, GPx, and CAT were determined using SOD (Nanjing Jiancheng Bioengineering Institute, Nanjing, China), GPx, and CAT (Beyotime Biotech, Shanghai, China) Antioxidant Enzyme Activity Assay Kits. The detailed experimental procedures can be found in Appendix A.

### 2.11. RNS Levels and Growth Curves of Yeast during Chronological Aging

YOM36 yeast was cultured in YPD overnight. Following a shift to synthetic defined (SD) medium and treatment with 1 µM RAP or 0, 0.1, 1, or 10 µM CYC (referred to as day 0), the cultured yeast at an initial OD_600_ value of 0.01 was released. The OD_600_ values of the YOM36 yeasts were measured using an Eppendorf Biophotometer Plus (Eppendorf Company, Hamburg, Germany) to construct a growth curve after incubation for 1, 2, 3.5, 5, 7, 9, 11, and 13 days. YOM36 yeasts were collected and broken down using a grinder (Jingxin Industrial Development Co., Ltd., Shanghai, China) at 70 Hz for one minute after the addition of grinding beads. The cell lysates were centrifuged at 4 °C and 12,000× rpm for 10 min to obtain the supernatant, which was used as a protein sample for assessing the RNS level according to the instructions provided with the RNS test kit (Bestbio Biotechnology Company, Nanjing, China).

### 2.12. Visualization of Autophagy

In summary, YOM38 yeast cells containing the pRS316-*GFP-ATG8* plasmid were initially cultured in YPD media for 24 h under gentle stirring in the dark. The yeast cells were then washed three times with PBS and transferred to SD media. Next, 20 mL of SD medium containing 300 µM RES or various concentrations (0, 0.1, 1, and 10 µM) of CYC was added to the yeast culture, which had an initial OD_600_ value of 0.1. The cells were incubated in the dark with shaking for 24 h. The SD medium was subsequently removed by performing three washes with PBS. DAPI staining was carried out at a final concentration of 1 µg/mL for 8 min under dark conditions. After staining, the cells were washed three times with PBS. Finally, the cells were suspended in a 30% glycerol solution, and the autophagy-induced yeast cells were visualized and imaged using a two-photon confocal fluorescence microscope (Olympus FV1000BX-51, Tokyo, Japan).

### 2.13. Western Blot Analysis

Western blot analysis was conducted according to the methodology described in a previous study [32]. In the time-course experiment, YOM38 yeast (pRS316-*GFP-ATG8* plasmid) was treated with 1 µM CYC for 0, 8, 16, 24, or 32 h. The observation point was chosen as 24 h. In the dose-course experiment, YOM38 yeast was treated with 0, 0.1, 1, or 10 µM CYC for 24 h. Additionally, 3T3 cells were treated with 10 µM AST or 0, 0.1, 1, or 10 µM CYC for 48 h. Protein was extracted from harvested cells from all treatment groups, and the protein concentration was measured using a BCA kit (CoWin Biotech, Beijing, China).

In brief, approximately 20 µg of protein from each sample was electrophoresed on sodium dodecyl sulfate polyacrylamide gels at 80 V for 15 min and then at 120 V for 60 min. The proteins were transferred to PVDF membranes (Bio-Rad Laboratories Inc., Hercules, CA, USA) and blocked for 90 min at room temperature with 5% nonfat dry milk buffer. The membranes were incubated with primary antibodies overnight at 4 °C, followed by incubation with secondary antibodies for 45 min at room temperature to visualize the protein bands. The protein bands were visualized using an enhanced chemiluminescence (ECL) Western blot kit (CoWin Biotech, Beijing, China), and the band density was quantified with ImageJ software (Version 1.42q, National Institutes of Health, Rockville, MD, USA). The primary and secondary antibodies used are listed in Appendix A.

### 2.14. Statistical Analysis

The analysis of the experimental data was performed using GraphPad Prism 8.0.2 software (GraphPad Software, San Diego, CA, USA). To determine significant differences between groups, one-way ANOVA followed by Tukey’s multiple comparison test was utilized. For the chronological lifespan data of yeast, the log-rank (Mantel–Cox) test was used. Each experiment was independently repeated three times, and the results are presented as the mean ± SEM. *p* < 0.05 indicates a significant difference compared to the negative control group.

## 3. Results

### 3.1. CYC Extends the Lifespan of Yeast and Inhibits the ETO-Induced Senescence of Mammalian Cells

K6001 yeast is an ideal candidate for screening compounds with anti-aging properties [22]. In this study, we isolated and purified multiple volatile compounds from *L. angustifolia* Mill., among which CYC demonstrated the most remarkable efficacy in extending the RLS of K6001 yeast. The structure of CYC is depicted in Figure 1a. Subsequently, we evaluated the anti-aging potential of CYC at doses of 0, 0.03, 0.1, 1, 3, 10, and 30 µM. To validate the reliability of our findings, we used RES as a positive control [14]. Among the selected concentrations, CYC could prolong the RLS of K6001 yeasts at 0.1 µM (*p* < 0.05), 1 µM (*p* < 0.01), 3 µM (*p* < 0.01), and 10 µM (*p* < 0.001), as shown in Appendix A. To easily distinguish the change in RLS after giving CYC and clearly display the results for subsequent mechanistic experiments, we only selectively showed the results of CYC at doses of 0.1, 1, and 10 µM (Figure 1b). Furthermore, we developed the prototrophic strain YOM36 to examine the effect of CYC on the CLS of yeast [33]. Compared with the negative control, 1 (*p* < 0.01) and 10 µM CYC (*p* < 0.001) notably improved the survival rates of yeast (Figure 1c). In conclusion, CYC extended both the RLS of K6001 yeast and the CLS of YOM36 yeast, indicating that CYC has significant anti-aging effects on yeasts.

To further validate the anti-aging effects of CYC, we employed a more reliable mammalian cell bioactivity testing system (i.e., PC12 and 3T3 cell lines). SA-*β*-Gal is widely recognized as a senescence biomarker in human and rodent cells [34]. To establish senescence cell models, we utilized ETO, a chemical drug known to cause DNA damage [35]. In PC12 cells, a dose of 1 µM ETO resulted in an increase in the percentage of SA-*β*-Gal-positive cells from 15.87 ± 0.65% in the negative control to 46.04 ± 6.22% (*p* < 0.001), signifying the successful establishment of the aging cell model. After the addition of RAP and CYC to the senescent cell model, the percentages of SA-*β*-Gal-positive cells in each group changed as follows: 20.33 ± 1.57% (*p* < 0.001) in the 10 µM RAP-treated group and 29.07 ± 1.31% (*p* < 0.01), 21.06 ± 1.99% (*p* < 0.001), and 15.31 ± 2.04% (*p* < 0.001) in the 0.1, 1, and 10 μM CYC-treated groups, respectively (Figure 1d,e). Moreover, in 3T3 cells, a dose of 0.3 µM ETO increased the proportion of SA-*β*-Gal-positive cells from 3.13 ± 0.60% in the negative control to 69.66 ± 4.68% (*p* < 0.001), indicating that 3T3 cells displayed greater sensitivity to ETO than did PC12 cells. However, treatment with 1 (*p* < 0.05) or 10 μM CYC (*p* < 0.001) or 10 μM TAU (*p* < 0.001) significantly reduced the SA-*β*-Gal content in 3T3 cells (Figure 1f,g). In summary, these findings collectively suggest that CYC has remarkable anti-aging effects on two distinct types of mammalian cells.

### 3.2. CYC Increases the Telomerase Content of Yeast and Elongates the Telomere Length of Mammalian Cells

Telomerase activity and telomere length are closely associated with longevity [4,5,6,7,8]. In this study, we investigated the effect of CYC on telomerase and the telomere length in both yeast and mammalian cells. The telomerase content in the negative control group was 22.07 ± 3.06 U/L, whereas that in the positive control group was 57.33 ± 5.12 U/L (*p* < 0.001). The 0.1, 1, and 10 µM CYC treatment groups exhibited telomerase levels of 45.57 ± 2.89 U/L (*p* < 0.01), 59.46 ± 5.03 U/L (*p* < 0.001), and 69.63 ± 5.70 U/L (*p* < 0.001), respectively (Figure 2a). Additionally, we investigated the impact of CYC on the telomere length in 3T3 cells to further validate our findings. The doses of 1 and 10 µM CYC increased the relative telomere length by 60.13% and 80.42%, respectively, compared to that in the negative control group (Figure 2b). These results indicate that CYC activates telomerase and extends the telomere length.

To explore the signaling pathways through which CYC protects telomeres, we examined its effects on shelterin proteins such as TRF2 and RAP1 [4,7,8]. Western blot assays were conducted to assess changes in TRF2 and RAP1 protein expression in 3T3 cells following treatment with AST and CYC. As shown in Figure 2c,d, CYC significantly enhanced TRF2 expression at concentrations of 0.1 (*p* < 0.001), 1 (*p* < 0.001), and 10 µM (*p* < 0.001). Moreover, treatment with 0.1, 1, or 10 µM CYC increased RAP1 protein expression by 7.61% (*p* < 0.05), 13.22% (*p* < 0.01), or 13.71% (*p* < 0.001), respectively, compared to that in the negative control group (Figure 2e,f). The complete unedited gel for Western blot analysis shown in Figure 2c,e is displayed in Appendix A. To summarize, these findings suggest that CYC exerts a protective effect on telomerase and telomeres by modulating the expression of TRF2 and RAP1.

The objective of this study was to investigate whether CYC can protect telomeres, even under stressful conditions. Based on the results shown in Figure 2g, we selected a dose of 6.5 µM H_2_O_2_ to establish an oxidative stress cell model. The telomere length of H_2_O_2_-treated 3T3 cells decreased by approximately 30% compared to that of the control cells (*p* < 0.05), confirming the damaging impact of oxidative stress on telomeres. The changes in the telomere length extension in the H_2_O_2_-induced group were 73.69% for the group treated with 10 µM AST, and 85.19% (*p* < 0.001) and 43.79% (*p* < 0.01) for the groups treated with 1 and 10 μM CYC, respectively (Figure 2h). Notably, compared with the control, 1 µM CYC reversed the telomere damage and significantly increased the telomere length by 50.17% (*p* < 0.001) (Figure 2h). In conclusion, these results suggest an interaction between oxidative stress and telomeres and indicate that CYC may play a role in regulating the cellular oxidative balance to mitigate the telomere damage caused by oxidative stress.

### 3.3. CYC Enhanced the Adaptation of Yeast under Oxidative Stress

Our findings, as shown in Figure 2h, demonstrate that CYC exerts a regulatory influence on oxidative stress. Therefore, we assessed the survival situations of BY4741 yeast under H_2_O_2_ stimulation to investigate whether the anti-aging mechanism of CYC involves antioxidative stress. The qualitative and quantitative results of yeast treated with 0.1, 1, and 10 µM CYC are presented in Figure 3a,b. Importantly, CYC enhanced yeast growth on a YPD plate containing 10 mM H_2_O_2_. Furthermore, in the quantitative assay, the survival rates for each group were 55.30 ± 2.21% for the negative control group, 69.08 ± 2.05% (*p* < 0.01) for the group treated with 10 µM RES, and 66.04 ± 1.51% (*p* < 0.05), 81.15 ± 1.17% (*p* < 0.001), and 66.03 ± 3.74% (*p* < 0.05) for the groups treated with 0.1, 1, and 10 µM CYC, respectively (Figure 3b). These findings suggest that the beneficial effects of CYC on yeast lifespan may be attributed, at least in part, to its antioxidative stress activity.

To assess the antioxidative stress effect of CYC, we investigated its impact on the ROS and MDA levels. Figure 3c shows a significant reduction in the ROS levels following the CYC treatment. At concentrations of 0.1, 1, and 10 µM, the levels decreased from 1073.28 ± 84.62 to 630.68 ± 107.28 (*p* < 0.01), 443.04 ± 53.73 (*p* < 0.001), and 464.21 ± 177.60 (*p* < 0.001), respectively. Additionally, the positive control, RES, at 10 µM, led to a decrease in the ROS levels from 1073.3 ± 84.62 to 447.3 ± 52.61 (*p* < 0.001). Furthermore, as shown in Figure 3d, the MDA concentration decreased from 0.35 ± 0.01 to 0.29 ± 0.01 (*p* < 0.01), 0.28 ± 0.01 (*p* < 0.01), and 0.28 ± 0.01 nmol/mg protein (*p* < 0.01) following treatment with 10 µM RES and 1 and 10 µM CYC, respectively. These findings highlight the ability of CYC to effectively decrease the ROS and MDA levels in BY4741 yeast, underscoring the crucial role of antioxidative stress in its anti-aging effects.

Antioxidant enzymes play a significant role in combating cellular oxidative stress, protecting organisms from the harmful effects of reactive oxygen species, and reducing oxidative damage [13]. SOD converts superoxide radicals (O_2_^●−^) into H_2_O_2_ and O_2_, CAT converts H_2_O_2_ into H_2_O and O_2_, and GPx eliminates harmful lipid peroxides from cells [13]. Therefore, we examined the impact of various doses of CYC on the activities of the antioxidant enzymes SOD, CAT, and GPx in yeast cells. Our results, presented in Figure 3e–h, demonstrate that the CYC treatment increased the activities of the total SOD, CuZn-SOD, CAT, and GPx in yeast cells. These findings indicate that CYC mitigates oxidative stress by enhancing the activity of antioxidant enzymes and scavenging free radicals, thereby contributing to the delay of aging processes.

### 3.4. Involvement of SOD1, SOD2, CAT, and GPx Genes in the Anti-Aging Effect of CYC

To further elucidate the association between the antioxidative stress effect of CYC and its anti-aging effect, we investigated the involvement of key genes, namely, SOD1, SOD2, CAT, and GPx, in the anti-aging effect of CYC. This investigation was carried out by assessing the replicative lifespans of relevant yeast mutants with a K6001 background. Cu/Zn-SOD (SOD1) and Mn-SOD (SOD2) are the two major enzymes of SOD [12,13]. RES can upregulate the expression of the SOD, CAT, and GPx genes and increase the activity of antioxidant enzymes, thereby reducing intracellular free radical accumulation and prolonging lifespan [14]. CYC extended the replicative lifespan of K6001 yeast. However, the longevity of the Δ*sod1*, Δ*sod2*, Δ*cat*, and Δ*gpx* yeast mutants, which closely resembled that of the corresponding control group (Figure 4a–d), was not prolonged by CYC. Similar outcomes were observed with RES treatment, suggesting a potential mechanistic similarity between CYC and RES [14]. These findings highlight the involvement of the SOD1, SOD2, CAT, and GPx genes in mediating the anti-aging effect of CYC.

### 3.5. CYC Decreases RNS Levels to Extend the Chronological Lifespan of Yeasts

RNS, such as NO, NO_2_, and ONOO^−^, play crucial roles in cell signaling and REDOX metabolism. However, excessive RNS production can induce nitration and nitrosylation reactions, altering the protein structure and impairing normal function [12]. Figure 1c shows that CYC could prolong the CLS of yeast. To investigate whether this effect is mediated through nitrosative stress, we monitored changes in the RNS levels at various growth phases during chronological aging. As shown in Figure 5a, both the CYC-treated and control groups exhibited log-phase growth after the first day, whereas the RAP-treated group required two days to enter this phase. The growth rate of yeast slowed upon entering the stationary phase on day 5 in all groups.

Figure 5b–d reveal that the level of RNS during the log-growth phase remained unaffected by the CYC treatment compared to that in the control group, except for a decrease observed with 1 µM CYC on the fifth day. In contrast, the RNS level significantly increased in the RAP-treated group in the log-growth phase from the second to the fifth day. Furthermore, CYC significantly reduced the RNS levels compared to those in the control group at 0.1 (*p* < 0.001), 1 (*p* < 0.001), and 10 μM (*p* < 0.001) upon entering the stationary phase on the seventh, ninth, and eleventh days (Figure 2e–g). Additionally, as depicted in Figure 5h, by day 13, the effect of CYC on the RNS levels was almost abolished, while RAP continued to significantly reduce the RNS levels (*p* < 0.001). This observation aligns with the CLS data (Figure 1c): the median survival of the RAP-treated group was 13 days, which was significantly longer than that of the CYC-treated group (11 days). These findings suggest that CYC decreases the RNS levels in the early stationary phase during chronological aging, thereby extending the CLS of yeast.

### 3.6. CYC Enhances Yeast Autophagy

Given the close association between autophagy and aging, we investigated the potential connection between autophagy and the anti-aging effect of CYC. *ATG2* and *ATG32* are key autophagy-related genes, and the knockout of *ATG2* in *Drosophila* and of *ATG32* in *Candida glabrata* leads to shortened lifespans [36,37]. As shown in Figure 6a,b, we observed that CYC failed to extend the lifespan of these mutants, indicating the involvement of autophagy. We then examined the levels of autophagy in the yeast treated with CYC using YOM38 yeast expressing the GFP-ATG8 fusion protein. GFP-ATG8 is a unique fusion protein that undergoes hydrolysis by vacuolar proteases during autophagy, releasing free GFP. Thus, the level of autophagy can be measured by the amount of free GFP [38]. Figure 6c,d indicated that the CYC significantly increased the percentage of cells with free GFP from 10.14 ± 0.46% to 14.91 ± 1.33% (*p* <  0.01), 20.95 ± 0.84% (*p*  <  0.001), and 16.22 ± 0.65% (*p*  <  0.001) following treatment with CYC at 0.1, 1, and 10 µM, respectively.

To further assess the impact of CYC on autophagy, Western blot experiments were conducted at various time points (0, 8, 16, 24, and 32 h) after the CYC treatment. Figure 6e,f shows that autophagy initiation occurred at 0 h after the CYC treatment, peaked at 24 h (*p* < 0.001), and returned to baseline levels by 32 h. Therefore, we selected 24 h as the time point for the dose–response experiments. Figure 6g,h show that the CYC treatment significantly increased the expression levels of GFP in yeast at concentrations of 0.1 (*p*  <  0.001), 1 (*p*  <  0.001), and 10 µM (*p*  <  0.001) CYC, respectively. The complete unedited gel for Western blot analysis shown in Figure 6e,g can be found in Appendix A. These findings indicate that CYC enhances the autophagy levels, potentially contributing to its anti-aging effects.

## 4. Discussion

In this study, we isolated and identified an anti-aging active compound called *β*-cyclocitral from *L. angustifolia* Mill. under the guidance of a K6001 yeast bioactive system. Our results showed that CYC prolonged the RLS and CLS of the yeast strains and reduced the level of the aging marker SA-*β*-Gal in ETO-induced PC12 and 3T3 cells, as shown in Figure 1. These findings verified that CYC has anti-aging effects on yeasts and mammalian cells. A recent study has shown that CYC is primarily produced when reactive oxygen species in plants oxidize *β*-carotene [16]. CYC has been identified as a novel plant signal that initiates stress tolerance and detoxification in plants [16,39]. Additionally, CYC has been found to act as a regulator of root stem cells, increasing lateral root branching and root meristem cell division [40]. In vitro and in vivo investigations have also revealed that CYC is a novel acetylcholinesterase inhibitor [41]. These studies provide evidence for further research on the anti-aging effects of CYC in animal models.

In our previous studies, we focused on analyzing the anti-aging mechanism of molecules through oxidative stress and autophagy [32]. In this study, we focused on telomerase and telomeres to undertake the research work. The changes in telomerase content, telomere length, and the expression of shelterin proteins TRF2 and RAP1 in Figure 2a–f suggested that CYC has anti-aging effects through the modification of telomeres. To investigate the interaction between telomeres and oxidative stress, we examined the impact of CYC on telomeres under oxidative stress conditions. The results in Figure 2g,h indicated that they had interaction, and CYC can rescue or even reverse the telomere damage induced by high concentrations of H_2_O_2_. Interestingly, low concentrations (0–6.0 mM) of H_2_O_2_ not only improved cell survival but also did not damage telomeres. This finding is consistent with previous research and may be attributed to the presence of a cellular self-regulation system that maintains a balance in cellular redox levels. During this process, antioxidant enzymes are activated, and their levels increase slightly compared to those in the control group [12,13,42,43].

Furthermore, we examined the impact of CYC on oxidative stress. The administration of CYC changed the survival rate of yeast under oxidative stress conditions, the activity of antioxidant enzymes, the levels of ROS and MDA in Figure 3, and the RLS of antioxidant gene mutants of yeast in Figure 4. These results clearly demonstrated that the antioxidative stress property of CYC plays a significant role in mitigating the aging process. Additionally, we investigated the effect of CYC on nitrosative stress in the CLS of yeast. Figure 5 provides evidence that nitrosative stress also plays a crucial role in the CLS of yeast and that the anti-nitrosative stress property of CYC contributes to its anti-aging effects.

In addition, we also investigated the effect of CYC on autophagy. The lack of impact of CYC on the RLS of the Δ*atg2* and Δ*atg32* yeast mutants, along with the significant increase in autophagic flux observed in the YOM38-*GFP-ATG8* yeast in Figure 6, indicate that autophagy also plays a vital role in the anti-aging effects of CYC.

CYC is an oxygen-containing monoterpene with unsaturated double bonds and aldehyde groups [16], which may contribute to its anti-aging activity. During our study, we found that the best concentrations of CYC for anti-aging and telomere were 10 μM, respectively. However, the optimal concentration of CYC was 1 μM in the antioxidant activity experiment. We considered that the reason may be the pro-oxidant properties of CYC at higher concentrations. A similar phenomenon occurs with the classic antioxidant vitamin C [44]. Vitamin C can act as either an antioxidant or a pro-oxidant under different conditions. Specifically, at high concentrations or in the presence of catalyti9c metals, vitamin C exhibits pro-oxidant properties [45]. CYC has similar chemical groups to vitamin C and may exhibit similar properties.

In our study, we utilized multiple positive substances to confirm the accuracy of the CYC experimental results on various models and mechanisms. RES, recognized as a classical antioxidant, exhibits robust scavenging activity against free radicals and stimulates antioxidant enzymes, thereby enhancing yeast survival under oxidative stress. These qualities qualify RES as a positive agent in such studies [14]. However, RES does not extend the chronological lifespan of yeast, in contrast to the efficacy demonstrated by RAP. RAP plays pivotal roles in cell growth, metabolism, autophagy, and apoptosis as an inhibitor of mechanistic targets of rapamycin. This contributes to its superior ability to extend the chronological lifespan compared to that of RES [14,18]. This superiority of RAP is attributed to its ability to mimic the effects of caloric restriction, a known mechanism of lifespan extension associated with mTOR inhibition [19,23]. Notably, RAP significantly extends the lifespan of yeast and attenuates aging in rat PC12 cells. Nevertheless, the cytotoxicity of RAP in mouse 3T3 cells precludes its suitability as a positive agent in this model, leading to its substitution with TAU due to its efficacy in attenuating ETO-induced senescence markers in 3T3 cells. Additionally, none of the tested substances (RES, RAP, or TAU) exhibited satisfactory effects on telomere function in 3T3 cells, prompting the selection of AST, the primary component of TA-65, a telomere-protective supplement [9]. These findings highlight the nuanced efficacy of anti-aging agents across diverse models and mechanisms, explaining the observed variations, such as the differential activities of CYC across different models.

Belsky and Kuo proposed that aging is a one-way chronological process that cannot be genuinely “reversed,” but its progression can vary in speed or “pace” throughout an individual’s life [46,47]. The capacity to monitor the rate of aging over time has been assessed for markers of telomere and telomerase, oxidative stress, DNA methylation, inflammation, and a variety of composite measures combining clinical and functional measurements [46]. In this study, we only demonstrated the anti-aging effect of CYC at the cellular level, which needs further confirmation in animal models in vivo. Since the active molecule is a volatile compound, the administration method and dose in animal models need to be carefully considered.

## 5. Conclusions

*β*-cyclocitral from *L. angustifolia* Mill. is an active molecule that exerts anti-aging effects on yeasts and mammalian cells. It prolonged the yeast lifespan of and decreased the SA-*β*-Gal content in mammalian cells via the modification of telomeres, antioxidative stress, and anti-nitrosative stress, as well as the enhancement of autophagy (Figure 7). This work laid the foundation for the profundity development of *L. angustifolia* Mill. In the future, we will perform a pharmaceutical evaluation of this compound in an aging animal model and clarify its mechanism of action.

## Figures and Tables

**Figure 1 antioxidants-13-00715-f001:**
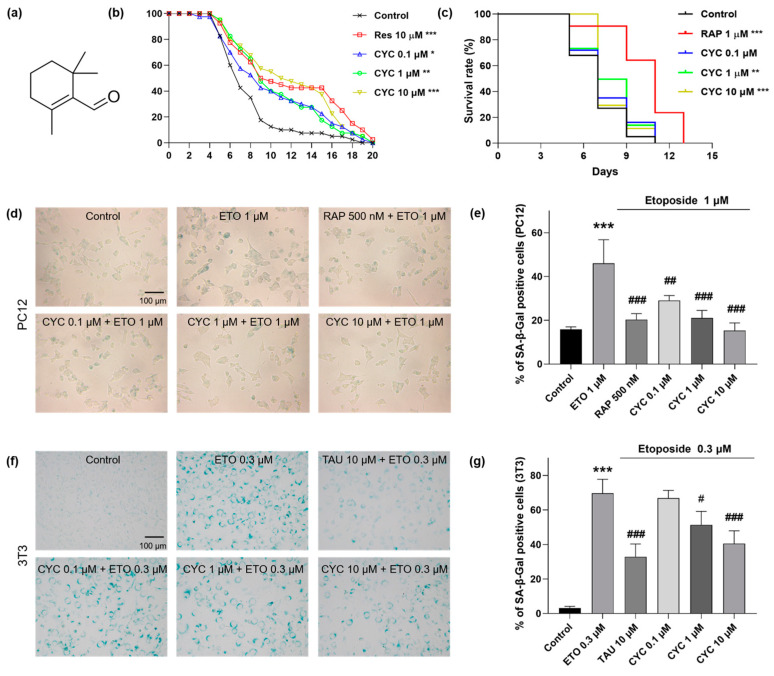
The molecule structure and anti-aging capacity of *β*-cyclocitral (CYC) in yeast and mammalian cells. (**a**) Chemical structure of CYC. (**b**) Effects of resveratrol (RES) at 10 µM or CYC at doses of 0.1, 1, and 10 µM on the replicative lifespan (RLS) of K6001 yeast. (**c**) Effects of rapamycin (RAP) at 1 µM or CYC at doses of 0.1, 1, and 10 µM on the chronological lifespan (CLS) of YOM36 yeast. (**d**,**f**) Anti-aging effect of CYC in etoposide (ETO)-induced senescence PC12 and 3T3 cells. (**e**,**g**) The digital result of (**d**,**f**). *, **, and *** represent significant differences at *p* < 0.05, *p* < 0.01, and *p* < 0.001 compared with negative control group, respectively. ^#, ##,^ and ^###^ represent significant differences at *p* < 0.05, *p* < 0.01, and *p* < 0.001 compared with ETO-induced group in, e.g., respectively.

**Figure 2 antioxidants-13-00715-f002:**
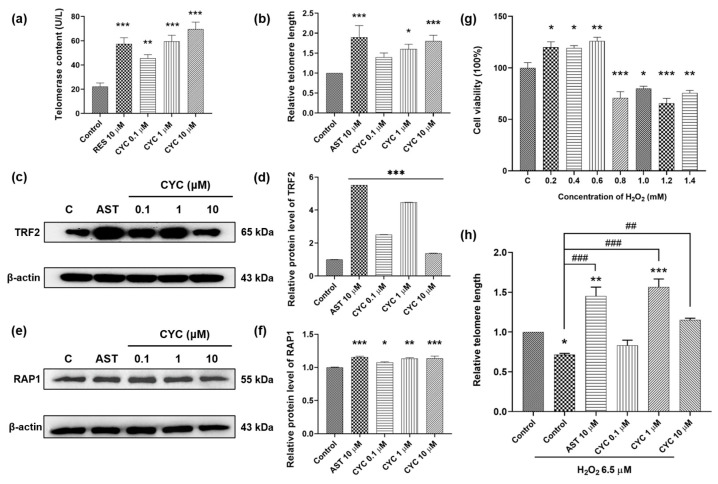
Effect of CYC on telomerase and telomeres in yeast and mammalian cells. (**a**) CYC increased telomerase contents in yeast. (**b**) Relative telomere length changed with 10 µM astragaloside IV (AST) or 0, 0.1, 1, and 10 µM CYC. The Western blot stripes of (**c**) telomeric-repeat binding factor 2 (TRF2) and (**e**) repressor activator protein 1 (RAP1) protein in 3T3 cells after incubation with 10 µM AST or 0, 0.1, 1, and 10 µM CYC. (**d**,**f**) The digital results of (**c**,**e**). (**g**) Effects of different concentrations of H_2_O_2_ on the cell viability of 3T3 cells. (**h**) Relative telomere length changed with 10 µM AST and 0, 0.1, 1, and 10 µM CYC applied 6.5 µM H_2_O_2_ compared with the control. *, **, and *** represent significant differences at *p* < 0.05, *p* < 0.01, and *p* < 0.001 compared with negative control group, respectively. ^##^ and ^###^ represent significant differences at *p* < 0.01 and *p* < 0.001 compared with 6.5 µM H_2_O_2_-induced control group in (**h**), respectively.

**Figure 3 antioxidants-13-00715-f003:**
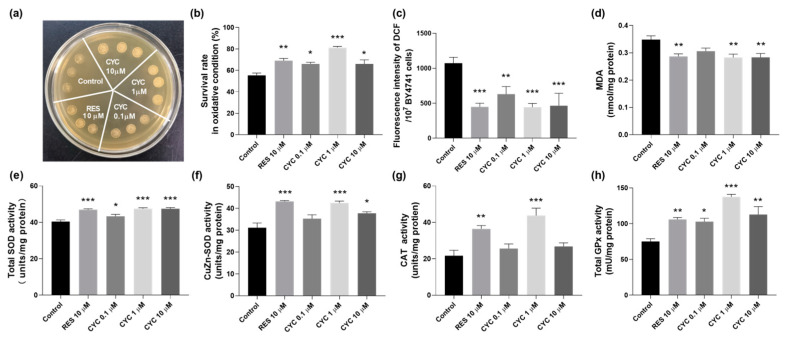
Effect of CYC on antioxidant capacity of BY4741 yeast. (**a**) Survival status of yeast cells under oxidative stress induced by 10 mM H_2_O_2_. (**b**) Quantification of yeast cell viability under 6.2 mM H_2_O_2_-induced oxidative stress. (**c**,**d**) reactive oxygen species (ROS) and malondialdehyde (MDA) levels and (**e**–**h**) total superoxide dismutase (SOD), CuZn-SOD, catalase (CAT), and glutathione peroxidase (GPx) antioxidant enzyme activities in yeast. *, **, and *** represent significant differences at *p* < 0.05, *p* < 0.01, and *p* < 0.001 compared with negative control group, respectively.

**Figure 4 antioxidants-13-00715-f004:**
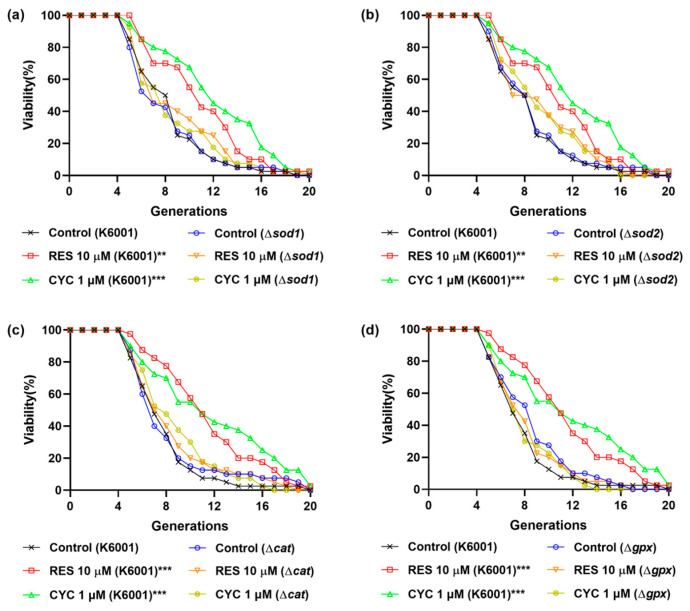
Effects of CYC on replicative lifespan of K6001 yeast and mutants related antioxidative enzyme genes. (**a**–**d**) The replicative lifespans of Δ*sod1*, Δ*sod2*, Δ*cat,* and Δ*gpx*. ** and *** represent significant differences at *p* < 0.01 and *p* < 0.001 compared with negative control group, respectively.

**Figure 5 antioxidants-13-00715-f005:**
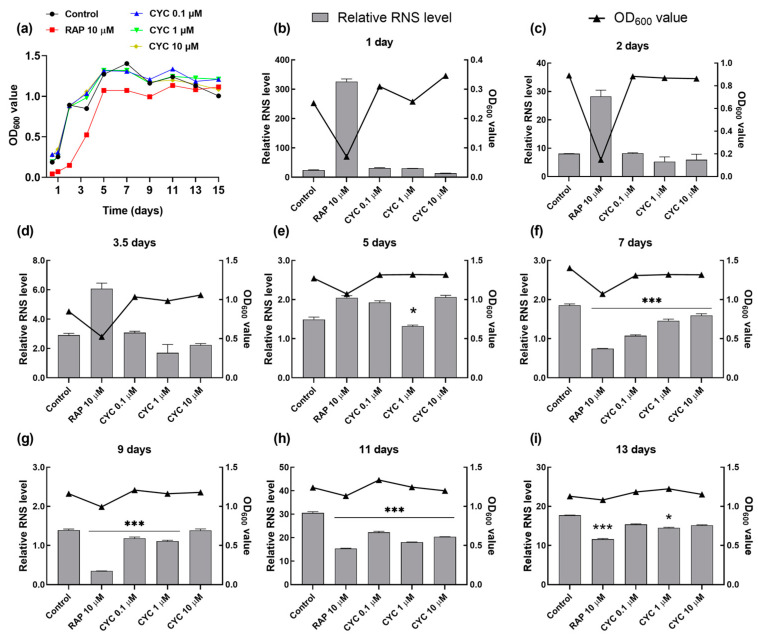
Effect of CYC on reactive nitrogen species (RNS) levels during chronological aging. (**a**) The growth curve of YOM36 yeasts during chronological aging. (**b**–**i**) The changes in the RNS levels after 10 μM RAP and 0, 0.1, 1, and 10 μM CYC treatment at 1, 2, 3.5, 5, 7, 9, 11, 13, and 15 days. * and *** represent significant differences at *p* < 0.05 and *p* < 0.001 compared with negative control group, respectively.

**Figure 6 antioxidants-13-00715-f006:**
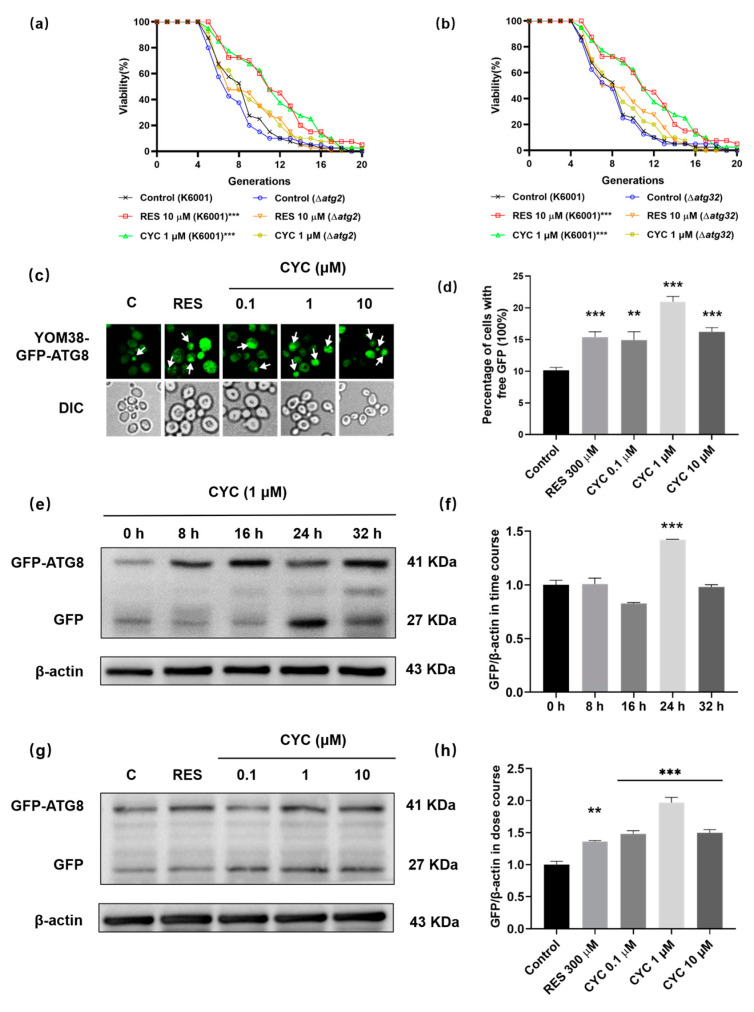
Effect of CYC on autophagy in yeast. (**a**,**b**) Failure of CYC to prolong the lifespans of Δ*atg2* and Δ*atg32* yeast mutants with K6001 background. (**c**) Fluorescent images of yeasts treated with 300 µM RES or 0, 0.1, 1, and 10 µM CYC. Punctate green fluorescence is free GFP representing the occurrence of autophagy. (**d**) Statistical result of (**c**). Ten pictures of each group was selected randomly, and the percentage of cells with free GFP in each group was calculated. (**e**) The Western blot results of GFP-ATG8 and free GFP in yeast after treatment with 300 µM RES or 1 µM CYC at 0, 8, 16, 24, and 32 h. (**f**) The digital result of (**e**). (**g**) The Western blot results of GFP-ATG8 and free GFP in yeast after treatment with 300 µM RES and 0, 0.1, 1, and 10 µM CYC for 24 h. (**h**) The digital result of (**g**). **, and *** represent significant differences at *p* < 0.01 and *p* < 0.001 compared with negative control group, respectively.

**Figure 7 antioxidants-13-00715-f007:**
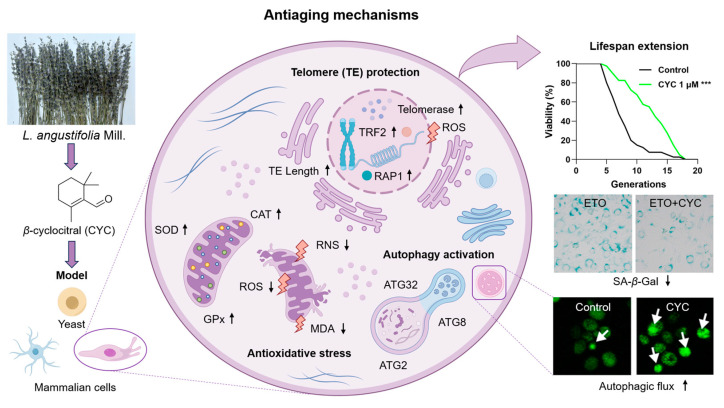
The proposed action mechanism of CYC. CYC plays an anti-aging role by activating telomerase and prolonging telomere length, antagonizing antioxidant stress, and inducing autophagy in yeast and mammalian cells, extending their lifespan. The black arrow ↑ represents the upregulation of antioxidant enzyme and telomeric protein expression, elongation of telomere length and activation of autophagic flux. The black arrow ↓ represents the downregulation of oxidation indicators (ROS, RNS, and MDA) levels. The purple arrows represent the research process. The white arrows are used to highlight the green fluorescent protein. ^***^ represent significant differences at *p* < 0.001 compared with the control group.

## Data Availability

All figures and data used to support this study are included within this article.

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
