# Peer review of "β-Cyclocitral from Lavandula angustifolia Mill. Exerts Anti-Aging Effects on Yeasts and Mammalian Cells via Telomere Protection, Antioxidative Stress, and Autophagy Activation"

_antioxidants, 2024, doi:10.3390/antiox13060715_

Round 1

Reviewer 1 Report

The authors used a replicative lifespan experiment of K6001 yeast to screen for anti-aging compounds within lavender extract (Lavandula angustifolia Mill.). This molecule had anti-aging effects on yeast and mammalian cells via the modification of telomeres, antioxidative stress, and anti-nitrosative stress, as well as the enhancement of autophagy. The quality of figures is commendable and the amount of data is adequate. However, there are some questions needed to be considered.

1. Introduction. The text is less logical, the connections between paragraphs need to be strengthened, the background and differences of this study should be explained in an orderly manner in relation to the current state of researches.

2. There are many abbreviations in the article, so the first time contains both the full name and abbreviation, you can just use the abbreviation directly afterward.

3. Material and methods: The methods should be characterized with a sufficient level of detail to allow the work to be reproduced. The text must report a minimum set of information to enable the reader to understand in what the method consisted of. The methods need to be described in combination with references.

(1) Line 139, there is no information on the plant material used fresh (wet), roasted, dried, by-product, etc. Is it directly used or there is pre-treatment?

(2) Line 164-168 and section 2.7., methods should be adequately described.

(3) Line 185. What are the steps of protein extraction?

(4) Line 190-191. What’s the volume of the sample supernatant, standard, and detection antibody?

(5) Line 240 and Line 244. 0.1, 1, and 10 µM.

4. Results. The language of the text is too cumbersome, please try to describe the figures as succinctly as possible.

(1) Line 309. LO?

(2) Line 312-313. What results does “these results” mean? Why did you choose 0.1, 1, and 10 µM CYC?

(3) Line 436-437. The results uniformly retained 2 digits after the decimal point.

5. Discussion.

(1) Much of the discussion of telomerase and telomeres, the antioxidative stress property, anti-nitrosative stress property, and autophagy is a repetition of results.

(2) The language lacks logic. Combine experimental data and the results of this study to elucidate the proposed action mechanism of β-cyclocitral clearly.

6. Conclusions: not a real conclusion, it is like the research contents. Figure 7. should be placed in “Discussion” for comprehensive analysis.

Reviewer 2 Report

Natural products are known to exert antioxidant effeetcs that might affect cell senescence in various ways. In this respect, your research is a valued contribution in this field. However, you should amend some of your claims as instructed below and also enrich the Discussion with more suggestions on CYC;s mode of action.

-Please reconsider your claims regarding the anti-aging properties of CTYC on the selected mammalian cells lines. In vitro cultured PC12 and 3T3 cells are far form resembling their normal counterparts in vivo and also the results of your study (as, for example, shown in Fig. 1 (b)) are not that spectracularly better tthan using other agents, e.g. resveratrol. I suggest that you chnage this claim in the title as "...and possibly mammalian cells".

-As also mentioned above, it is not clear that CYC has a considerably better performance as an antiaging compound compared to other established antioxidants, such as rerveratrol. Also, it is not adequatedly explained why the positive antioxidant effect of CYC is reduced at concentrations higher than 1 μM (see Fig. 3). Is there a possibility of a pro-oxidant property at higher conentrations? The authors should offer some hypothesis on this aspect.

- Line 552: It is impossible to claim a universal applicability of CYC since you have tested just two mammalian cell lines, please omit this claim.
